# Enhanced Photodegradation Stability in Poly(butylene adipate-*co*-terephthalate) Composites Using Organically Modified Layered Zinc Phenylphosphonate

**DOI:** 10.3390/polym12091968

**Published:** 2020-08-30

**Authors:** Jie-Mao Wang, Hao Wang, Erh-Chiang Chen, Yun-Ju Chen, Tzong-Ming Wu

**Affiliations:** Department of Materials Science and Engineering, National Chung Hsing University, 250 Kuo Kuang Road, Taichung 402, Taiwan; D9866023@mail.nchu.edu.tw (J.-M.W.); eddie115923@icloud.com (H.W.); erchiang.chen@gmail.com (E.-C.C.); bitcoin2243@gmail.com (Y.-J.C.)

**Keywords:** photodegradation, biodegradable polymer, PBAT, PPZn

## Abstract

The enhancement of the ultraviolet (UV) photodegradation resistance of biodegradable polymers can improve their application efficacy in a natural environment. In this study, the hexadecylamine modified layered zinc phenylphosphonate (m-PPZn) was used as a UV protection additive for poly(butylene adipate-*co*-terephthalate) (PBAT) via solution mixing. The results from the Fourier transform infrared spectroscopy (FTIR) and wide-angle X-ray diffraction analysis of the m-PPZn indicated the occurrence of hexadecylamine intercalation. FTIR and gel permeation chromatography were used to characterize the evolution of the PBAT/m-PPZn composites after being artificially irradiated via a light source. The various functional groups produced via photodegradation were analyzed to illustrate the enhanced UV protection ability of m-PPZn in the composite materials. From the appearance, the yellowness index of the PBAT/m-PPZn composite materials was significantly lower than that of the pure PBAT matrix due to photodegradation. These results were confirmed by the molecular weight reduction in PBAT with increasing m-PPZn content, possibly due to the UV photon energy reflection by the m-PPZn. This study presents a novel approach of improving the UV photodegradation of a biodegradable polymer using an organically modified layered zinc phenylphosphonate composite.

## 1. Introduction

Most nonbiodegradable plastic wastes cause significant land and marine pollution [1]. This problem has changed the direction of serious attention towards ecological sustainability and the possible impact on related animals and plants [2,3,4]. The application of biodegradable polymers is considered to be one among the effective methods in alleviating the problem at hand, especially in agricultural applications, because of their recyclability [5,6,7]. However, in agricultural applications, it is important to consider the photodegradation of polymers by ultraviolet (UV) light [8]. Two basic types of ultraviolet rays reach the Earth’s surface—UV-B (280–320 nm) and UV-A (320–400 nm)—that cause polymer chain scission and molecular weight reduction while affecting the function of the material. It is worth noting that the UV-C (100–280 nm) of solar radiation is absorbed by the Earth’s atmosphere [9,10]. Thus, some organic or inorganic additives have been used to improve the stability of polymers with UV irradiation [11,12,13,14].

Poly(butylene adipate-*co*-terephthalate) (PBAT), one of the most important biodegradable copolymers, is synthesized using 1,4-butanediol, adipic acid, and terephthalic acid or dimethylene terephthalate [15,16]. PBAT is a commonly used polymer in agriculture, especially in the application of mulch films [17,18,19]. PBAT wastes are nontoxic when biodegraded in the soil, which conforms to the ecotoxicity requirement of the ASTM standard [20,21]. To reduce the UV photodegradation of PBAT, different additives have been studied, such as montmorillonite (MMT), carbon black, and SiO_2_ [13,22,23]. UV protection can be achieved by the application of additives that can absorb or reflect photon energy, thus reducing the photodegradation of the polymer matrix in polymer-based composite materials [13]. Zinc phenylphosphonate (PPZn, Zn(O_3_PC_6_H_5_)···H_2_O), a two-dimensional layered material, has received significant attention due to its ability to improve crystallization, mechanical properties, and biodegradation in various polymers [24,25,26]. Since PPZn shows broad UV absorption bands between 300 and 400 nm, we are concerned with their effect on the UV photodegradation of biodegradable polymer/PPZn composites [27]. This study is the first to discuss the feasibility of PPZn to affect the photodegradation of PBAT. The loss of a coordinated water molecule leaves an open coordination site on the zinc atom for possible intercalation or catalytic reactions [28,29]. Thus, for better compatibility between PPZn and the polymer matrix, an organic modifier based on a long carbon chain was used [30,31].

In this study, a long-chain alkylamine, hexadecylamine, was used to produce an organically modified PPZn (m-PPZn). PBAT and m-PPZn were mechanically mixed in a dichloromethane solution to prepare PBAT/m-PPZn nanocomposite materials. This study aims to investigate the UV protection effect of m-PPZn on PBAT. An artificial light source was used to induce the photodegradation of the PBAT/m-PPZn samples. The molecular weight and delta yellowness index of samples were used to evaluate the UV protection characteristics of m-PPZn in the PBAT matrix.

## 2. Materials and Methods

### 2.1. Materials

PBAT (Ecoflex^®^ C1200) was supplied by BASF (Ludwigshafen, Germany). The chemicals used for the synthesis and modification of m-PPZn were phenylphosphonic acid (C_6_H_5_P(O)(OH)_2_), zinc nitrate hexahydrate (Zn(NO_3_)_2_···6H_2_O), sodium hydroxide (NaOH), and hexadecylamine (CH_3_(CH_2_)_15_NH_2_), which were acquired from Sigma-Aldrich (St. Louis, MO, USA) without further purification.

### 2.2. Synthesis and Modification of m-PPZn

PPZn was prepared by a similar approach reported previously [25]. Typically, 1 g phenylphosphonic acid and 1.88 g zinc nitrate hexahydrate were individually dissolved in 40 and 20 mL, respectively, of deionized water, then mixed together, followed by the addition of 0.1 M aqueous sodium hydroxide to reach pH 5–6. The solution was stirred at 60 °C for 3 days, and after this, the obtained insoluble (i.e., PPZn) was filtered, washed, and dried at 60 °C under vacuum for 3 days. Modification with hexadecylamine was carried by mixing PPZn and hexadecylamine in ethanol solution at room temperature for 7 days. The obtained m-PPZn was filtered and dried at 60 °C under vacuum for 3 days.

### 2.3. PBAT/m-PPZn Composites Preparation

PBAT and m-PPZn were separately dissolved in dichloromethane for 1 h, then mixed for another 1 h. The mixed solution was dried in vacuum at 40 °C for 3 days. The obtained composites were identified as PBAT/m-PPZn01, PBAT/m-PPZn05, and PBAT/m-PPZn10, which represented the 0.1, 0.5, and 1 wt % of m-PPZn in composites, respectively. These composites, and neat PBAT, were hot pressed using a hot press machine under the pressure of 10 kgf/cm^2^ at 120 °C for 10 minutes. The samples were cut into 1 × 4 cm^2^ pieces for further investigation.

### 2.4. Artificial Photodegradation Test

The samples were irradiated under a UV lamp (Philips CLEO HPA 400S, Amsterdam, The Netherlands) with a radiation between 300 and 400 nm. The irradiation times were 7, 14 and 21 days. The value of irradiance measured at the level of the sample surface was 6 mJ/cm^2^, and the dose used during 1-day exposure was 518.4 J/cm^2^. The temperature of the sample surface was about 45 °C. The relative humidity of the environment was about 50%.

### 2.5. Characterization

Fourier transform infrared spectroscopy (FTIR) experiments of PPZn, hexadecylamine, and m-PPZn were carried out on a Spectrum One spectrometer (Perkin-Elmer, Waltham, MA, USA) in the range from 450–4000 cm^−1^. The PBAT and its composites were performed in attenuated total reflection (ATR) mode.

UV-Vis spectra were acquired by a U-3900 UV-Vis spectrophotometer (Hitachi, Tokyo, Japan) in the range of wavelength from 250 to 400 nm. The optical reflectance of the m-PPZn was measured with an integrating sphere, and the spectra of the composites were acquired in transmittance mode.

The transmission electron microscopy (TEM) images of the composites were performed using JEM-1400 (JEOL, Tokyo, Japan). 

An X-ray diffractometer (Bruker D8, Karlsruhe, Germany) equipped with Ni-filtered Cu Kα radiation was used for the experiments of wide-angle X-ray diffraction (WAXD). The measurements of WAXD were carried out in the range of 2θ = 1–40° at a scanning rate of 1°/min. 

Molecular weights of the samples were determined by gel permeation chromatography (LC-4000, JASCO, Tokyo, Japan) with refractive index detectors (RI-4030, JASCO, Tokyo, Japan) calibrated with standard polystyrene. Dichloromethane was used as the mobile phase with a 1 ml/min flow rate.

The yellowness index (*YI*) of the sample before and after UV irradiation was measured using a spectrophotometer (CM-3600d, Konica Minolta, Osaka, Japan), which gave a direct *YI* value on the basis of the ASTM method D1925. The *YI* was represented in terms of the delta yellowness index (*dYI*), which defined as [32]:(1)dYI=YI after irradiation−YI before irradiation

## 3. Results and Discussion

The FTIR spectra of PPZn, m-PPZn, and hexadecylamine are shown in Figure 1. A close observation of the PPZn FTIR spectra shows a broad band at 1633 cm^−l^ and between 3400–3500 cm^−1^, which are related to the water molecules coordinated to Zn. The peaks at 650–750 cm^−1^ are attributed to the phenyl ring. The absorption peaks at 980–1200 cm^−1^ can be attributed to the PO_3_ group of phosphonic acid [29,31]. In the FTIR spectra of hexadecylamine, the peaks at 2922 and 2851 cm^−1^ can be attributed to the antisymmetric and symmetric effects, respectively, which is also shown in m-PPZn. Furthermore, the peaks at 1469 and 3335 cm^−1^ can be attributed to the –CH_3_ and –NH_2_ end groups [33,34]. Notably, in the spectrum of m-PPZn, the peaks of –NH_2_ shift to a lower wavenumber, 3301 cm^−1^, which suggests that the amine group of hexadecylamine was coordinated to the zinc atom [28,29].

Figure 2 shows the UV-Vis reflectance spectra of the PPZn and m-PPZn. In the UV-A and UV-B regions, more than 80% of the UV light is reflected, indicating the barrier ability of the PPZn. A lower reflectance in the UV-C region is attributed to the absorption via the π→π* transition of the phenylphosphonate anion [27]. In the region of UV-A and UV-B, the organic modifier hexadecylamine was slightly reduced in the reflectance of the PPZn.

Figure 3 shows the WAXD result of the m-PPZn, PBAT, and PBAT/m-PPZn composites. The data of m-PPZn were normalized to the peak at 2θ = 3.30°. The data of PBAT and the composites were normalized to the peak at 2θ = 23.21°. In our previous report, three diffraction peaks at 2θ = 6.34°, 12.49°, and 18.68° were observed for the PPZn, which correspond to the (010), (020), and (030) crystal planes, respectively [35]. In this study, three diffraction peaks at 2θ = 3.30°, 6.51°, and 9.72° were observed for the PPZn, which occur at the (010), (020), and (030) crystal planes, respectively. The interlayer spacing, *d*(010), of PPZn and m-PPZn were determined to be 1.39 and 2.67 nm, respectively, via Bragg’s equation. Including the result of FTIR and WAXD, this demonstrates that hexadecylamine was successfully intercalated into the interlayer spacing of PPZn. Five diffraction peaks at 2θ = 16.19°, 17.40°, 20.50°, 23.21°, and 25.12° were observed for PBAT, which were found to be consistent with those of the crystalline polybutylene terephthalate (PBT) [15]. By increasing the m-PPZn content, the peak positions and intensities of the m-PPZn-related peaks remained unchanged and increased, respectively. These results indicate that the initial stacking structure of the m-PPZn is retained in the PBAT/m-PPZn composites, and the addition of the m-PPZn does not change the crystal structure of the PBAT. Moreover, the morphology of PBAT/m-PPZn-10 was investigated using TEM, as shown in Figure 4. The stacking structure of m-PPZn is observed, which is consistent with the WAXD data.

Figure 5 shows the UV-Vis transmittance spectra of the PBAT and the PBAT/m-PPZn composite materials between 250–400 cm^−1^. The UV-Vis results of the PBAT material show excellent UV-A penetration characteristics; however, they show poor UV-B and UV-C penetration characteristics due to the presence of the benzene ring and carbonyl group [36,37]. The UV-Vis results of the PBAT/m-PPZn composites show better UV barrier properties than that of the PBAT. Furthermore, the UV-A and UV-B penetration characteristics are reduced as the m-PPZn loading increases. Considering that m-PPZn has good UV-A and UV-B reflectance properties, the transmittance of the PBAT/m-PPZn composites decreases with increasing m-PPZn, indicating that the photon energy reflection by m-PPZn might be the main mechanism.

Figure 6 shows the FTIR spectra of PBAT and the PBAT/m-PPZn composites before and after 21 days of UV irradiation. Before the UV irradiation, the FTIR spectra of PBAT and the PBAT/m-PPZn composites showed an absorbance peak at 1721 cm^−1^, which is attributed to the C=O group (Rc=o). The peaks at 1268, 1250, 1166, 1118, and 1103 cm^−1^ were attributed to the ester group. The peaks at 720 cm^−1^ were attributed to the methylene groups, while those at 1505 and 875 cm^−1^ were attributed to the benzene ring [38,39]. The intensities of these peaks became insignificant after irradiation, which indicated the chemical decomposition of the PBAT via photodegradation. In addition, a broad carbonyl peak (1850–1550 cm^−1^) was observed, which indicated the chain scission of the ester group. The left shoulder (1850–1750 cm^−1^) of the carbonyl peak indicates the formation of the free C=O (Fc=o), while the right shoulder (1670–1550 cm^−1^) indicates the formation of a lower molecular weight ester (LMWE) [10]. The chain scission at the ester linkages of the PBAT from the photodegradation was via the Norrish I process, since the carbonyl group is easily photodegraded [10,38]. 

Similarly, the FTIR spectroscopy of the PBAT/m-PPZn composites was also carried out. The FTIR result shows the area of the corresponding deconvolution peak, which can be used to evaluate the ratio of related chemical groups [40,41]. In the deconvolution process of the FTIR spectra between 1850–1550 cm^−1^, the peak close to 1780 cm^−1^ was assigned to F_C=O_, the peak close to 1721 cm^−1^ was assigned to *R*_C=O_, and the peak close to 1646 cm^−1^ was assigned to LMWE. The ratio of the area under the peak assigned to the F_C=O_ and *R*_C=O_ group (F_C=O_/*R*_C=O_) was used to determine the evolution of photodegradation. A high value of F_C=O_/*R*_C=O_ indicates more photodegradation in the PBAT. The FTIR deconvolution results of the PBAT after 21 days irradiation are shown in Figure 7, and the F_C=O_/*R*_C=O_ ratio is presented in Table 1. The F_C=O_/*R*_C=O_ ratio of the PBAT/m-PPZn composites is also shown in Table 1. The decrease in the F_C=O_/*R*_C=O_ value with increasing m-PPZn content demonstrates that the m-PPZn retards the photodegradation of the PBAT.

The appearance brought about by changes in color is another way to identify the differences in photodegradation. In Figure 8, PBAT and the PBAT/m-PPZn composites with different m-PPZn content show a clear discoloration after the UV irradiation. The samples changed to yellow after irradiation. Under irradiation, the exposed aromatic rings can be oxidized to phenoxy radicals, which cause the yellowing of the polymer due to their strong absorbance in the visible region. The yellowing phenomenon of the PBAT, which follows a similar mechanism, was reported [42,43,44]. The yellowing phenomenon is due to the oxidation of the benzene ring, and the *dYI* is used to determine the photodegradation of the PBA. The higher the *dYI*, the stronger the degradation. Figure 9 shows the *dYI* of the samples after UV irradiation. The *dYI* of each sample increases daily with the irradiation, which indicates the evolution of photodegradation. As the content of m-PPZn in the composite increases, the *dYI* of the sample decreases, which means that m-PPZn hinders the photodegradation of PBAT.

Figure 10 shows the average molecular weights (*M_n_*) of the PBAT and PBAT/m-PPZn composites after UV irradiation. The result shows a remarkable molecular chain scission in the first week, which reduces as the m-PPZn content increases. After 21 days of irradiation, the PBAT/PPZn-10 retains the highest molecular weight, showing the best resistance to photodegradation. The result indicates that m-PPZn can play a significant role in photodegradation protection. In a study conducted by Chen et al., it was reported that the MMT layers could contribute to the UV reflection ability, resulting in less photon energy interacting with the PBAT [13]. A similar approach was adopted in this study through the effect of a similar two-dimensional layered m-PPZn in the PBAT-m-PPZn composite materials. A common problem associated with the outdoor application of polymers is the chain scission, which reduces their performance [10]. In agricultural applications, such as the use of mulch films, it is pertinent to consider the photodegradation of polymers by ultraviolet (UV) light. Thus, this study provides a suitable method of increasing the resistance of PBAT to photodegradation.

## 4. Conclusions

In this study, the application of hexadecylamine modified PPZn to PBAT to enhance UV protection was demonstrated. WAXD results and TEM images of the PBAT/m-PPZn composites revealed that the m-PPZn retained its stacking structure in the composite materials. The UV barrier properties were significantly enhanced by the m-PPZn content in the composite materials. Through the characterizations carried out, the yellowness index and molecular weight observed, all experimental results indicated that the photodegradation of PBAT decreased with increasing m-PPZn loading. This degradation could be attributed to the excellent UV reflective ability of the m-PPZn. Thus, the findings in this study can be used to reduce the photodegradation of polymers exposed to sunlight, especially for agricultural use.

## Figures and Tables

**Figure 1 polymers-12-01968-f001:**
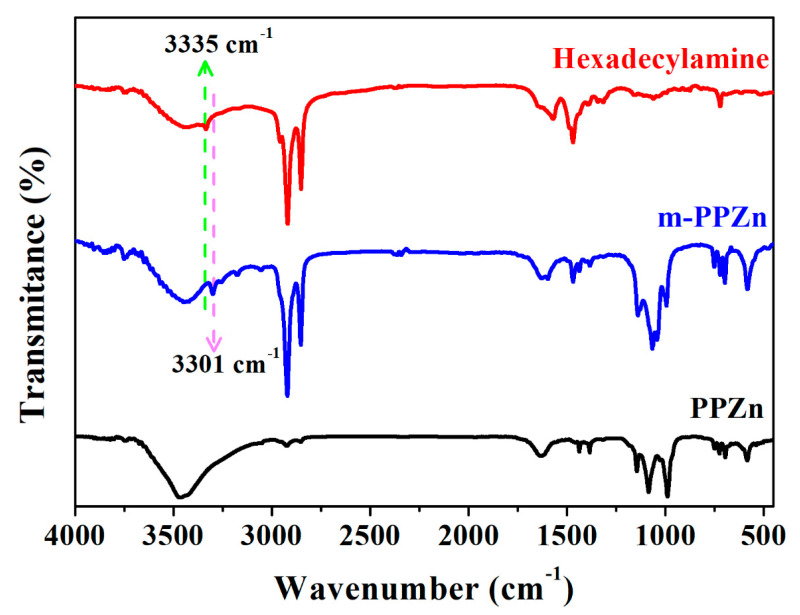
FTIR result of PPZn, m-PPZn, and hexadecylamine.

**Figure 2 polymers-12-01968-f002:**
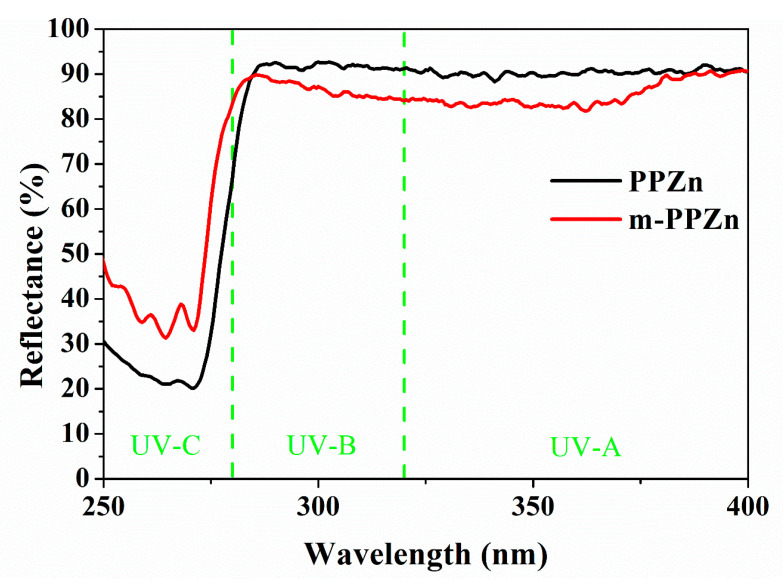
UV-Vis spectra of PPZn and m-PPZn.

**Figure 3 polymers-12-01968-f003:**
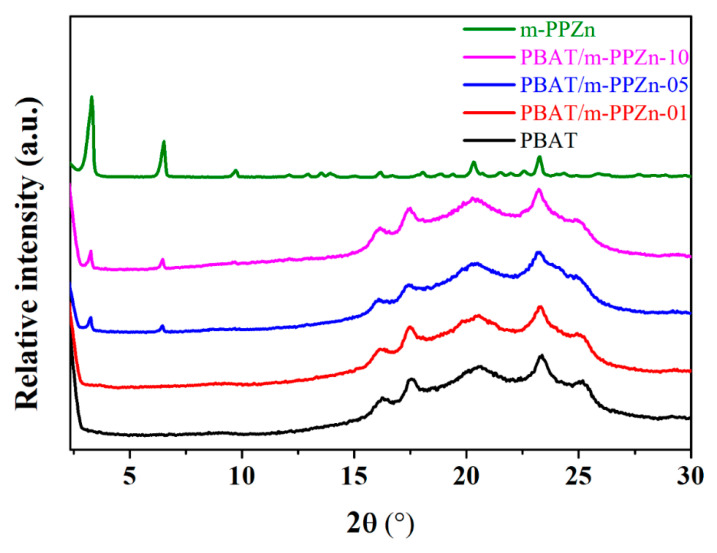
WAXD results of m-PPZn, PBAT, and PBAT/m-PPZn composites.

**Figure 4 polymers-12-01968-f004:**
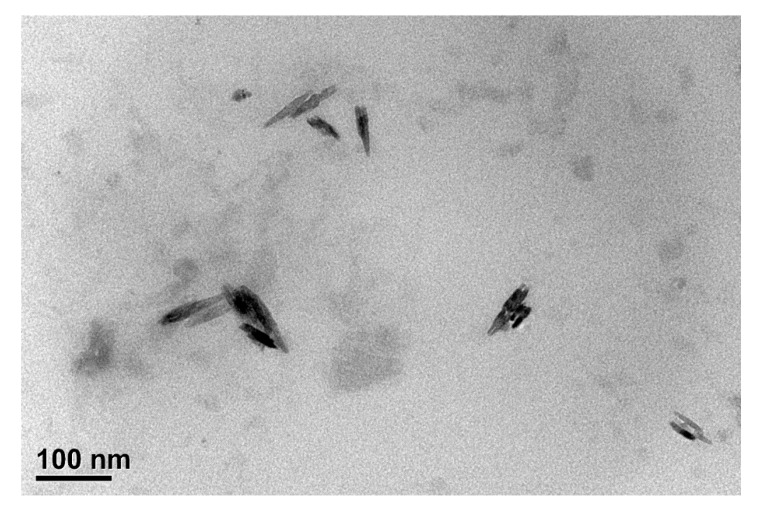
TEM image of PBAT/m-PPZn-10.

**Figure 5 polymers-12-01968-f005:**
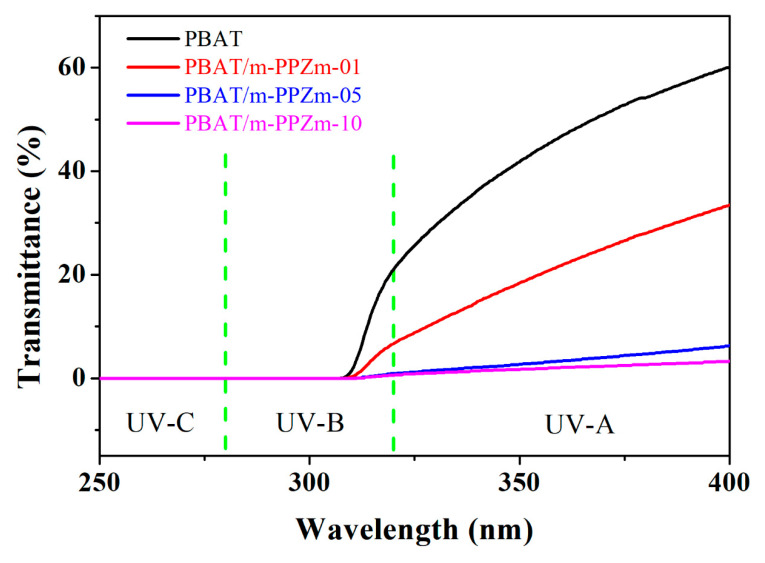
UV-vis result of PBAT and PBAT/m-PPZn composites.

**Figure 6 polymers-12-01968-f006:**
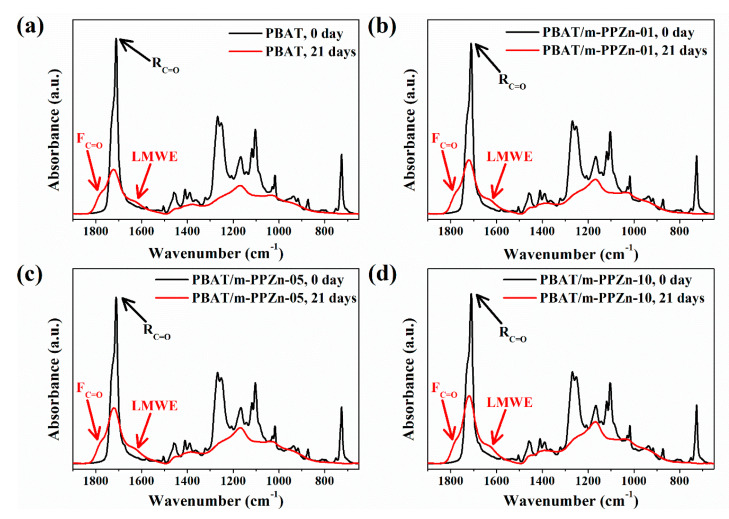
FTIR spectra of (**a**) PBAT, (**b**) PBAT/m-PPZn-01, (**c**) PBAT/m-PPZn-05, and (**d**) PBAT/m-PPZn-10 before and after 21 days UV irradiation.

**Figure 7 polymers-12-01968-f007:**
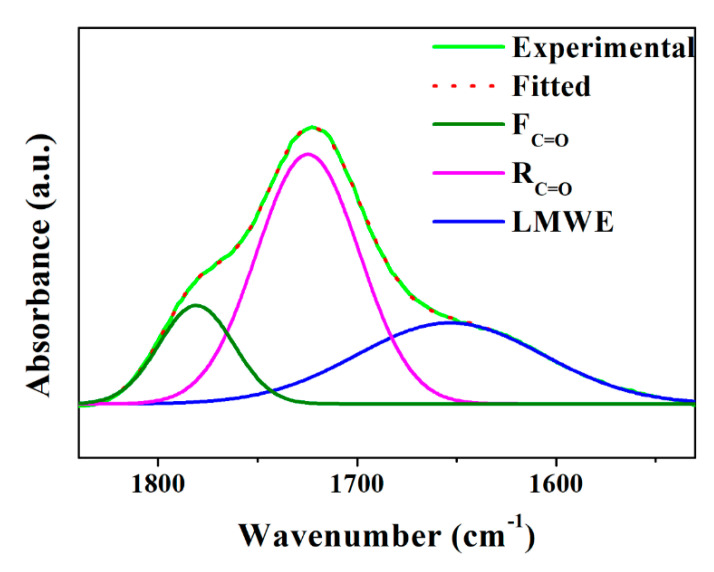
FTIR deconvolution spectrum of PBAT after 21 days UV irradiation.

**Figure 8 polymers-12-01968-f008:**
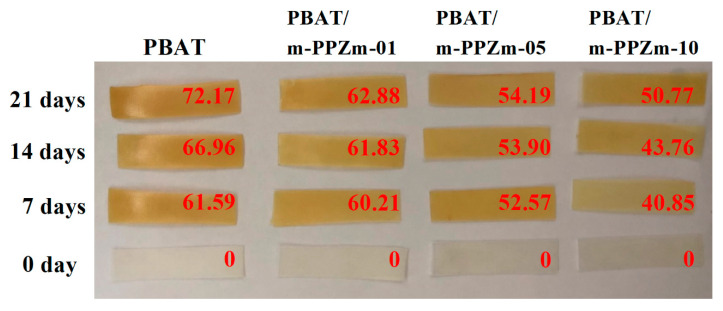
Photos of PBAT and PBAT/m-PPZn composites after 21 days of UV irradiation. The insert value (red) is *dYI* of each sample.

**Figure 9 polymers-12-01968-f009:**
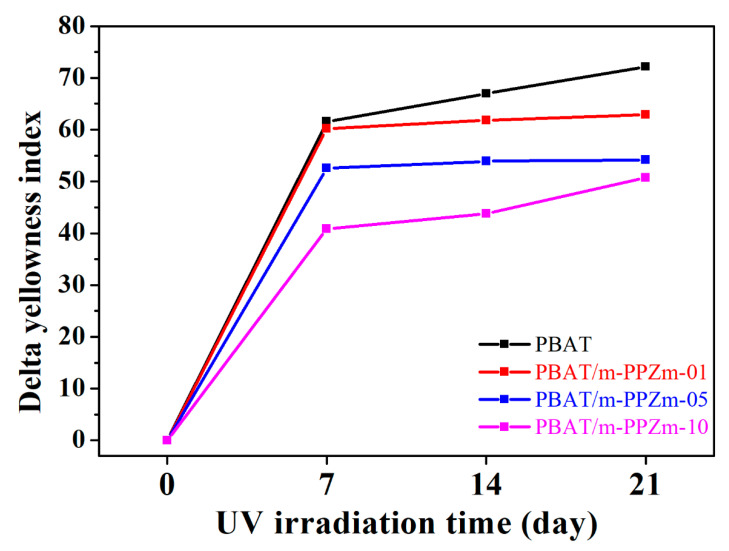
Delta yellowness index of PBAT and PBAT/m-PPZn composites after 21 days of UV irradiation.

**Figure 10 polymers-12-01968-f010:**
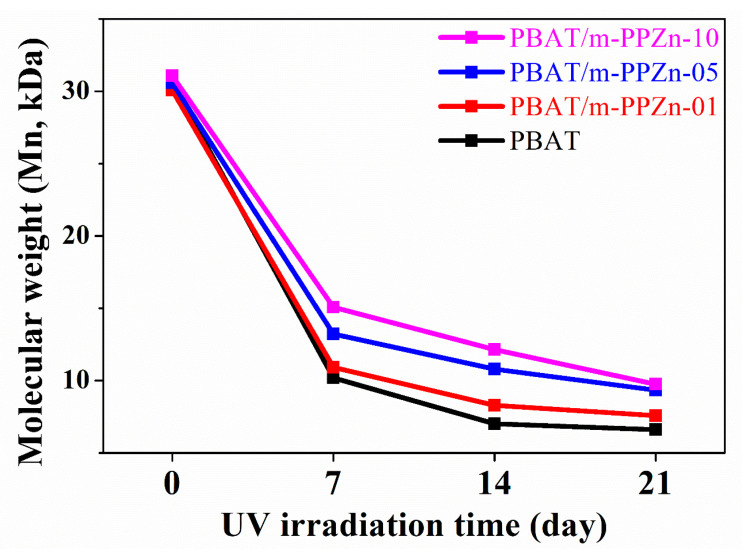
Average molecular weights (*M_n_*) of PBAT and the PBAT/m-PPZn composites after 21 days of UV irradiation.

**Table 1 polymers-12-01968-t001:** Ratio of F_C=O_ to *R*_C=O_ of samples after 21 days UV irradiation.

Sample Name	PBAT	PBAT/m-PPZn-01	PBAT/m-PPZn-05	PBAT/m-PPZn-10
$F_{C=O}/R_{C=O}$	0.286 ± 0.002	0.266 ± 0.002	0.246 ± 0.002	0.235 ± 0.003

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
