# Peer review of "Enhanced Photodegradation Stability in Poly(butylene adipate-co-terephthalate) Composites Using Organically Modified Layered Zinc Phenylphosphonate"

_polymers, 2020, doi:10.3390/polym12091968_

Round 1

Reviewer 1 Report

This manuscript presents the photodegradation of poly(butylene adipate-co-terephthalate) (PBAT) after the addition of organically modified layered zinc phenyl phosphonate (m-PPZn). The manuscript is not well-written and is difficult to follow (the language check I needed). Besides, the stability against the photodegradation according to the carbonyl index is too low and could be in the error level. I regret that this is not suitable for publication. There are some detailed concerns:

  1. Line 50; The authors stated that “Considering both PPZn and MMT are two-dimensional layered inorganic material, the feature on UV photodegradation of polymer/PPZn composites attract our attention.” This vague and weak motivation. The authors should motivate the study stronger in terms of literature review and studying underlying mechanisms.
  2. The FTIR spectra are needed to be clarified in the way how are they normalized to a characteristic peak? (Figure 1). If not, that would be misleading.
  3. The authors used relative intensity in the XRD Figure? Normalized to which peak?
  4. The authors stated that (Line 136); “Moreover, the morphologies of PBAT/m-PPZn-10 was investigated by TEM, shown in Figure 4. The stacking structure of m-PPZn is clearly shown, which is good agreement with the WAXD data.” The TEM doesn’t show this conclusion?!!
  5. Figure 6 together with is too misleading. The authors need to assign RC=O and FC=O clearly in Figure 6. Then the authors need to include one of the nanocomposites before and after irradiation in the Figure. The results in Table 1 showed that the ratio (indicating the photodegradation) for all samples is almost in the same range. However, the authors stated that (Line 170) “The decrease in value of ratio with increasing m-PPZn content demonstrates that m-PPZn retards the photodegradation of PBAT.”, which is not true.
  6. Figure 10 is also misleading. Which molecular weight? Mn or Mw? How are the PDI and molecular weight distribution? The authors need to bring all information at least in supporting information.
  7. The authors need to discuss the kinetics of photodegradation.

Author Response

Response to the comments of reviewers

The authors appreciate the referees’ careful reading and thoughtful suggestions. Points by point responses to reviewer’s comments are discussed below.

Reviewer #1:

This manuscript presents the photodegradation of poly(butylene adipate-co-terephthalate) (PBAT) after the addition of organically modified layered zinc phenyl phosphonate (m-PPZn). The manuscript is not well-written and is difficult to follow (the language check I needed). Besides, the stability against the photodegradation according to the carbonyl index is too low and could be in the error level. I regret that this is not suitable for publication.

Answer: The writing of this manuscript in English has been edited by Enago, Crimson Interactive Pvt. Ltd. The certificate of editing is attached below. The data of carbonyl index and experimental error has been newly added in Table 1. From this result, it is clear that the carbonyl index is decreased with increasing the m-PPZn contents.

Table 1. Ratio of FCO to RCO of samples after 21 days UV irradiation.

Sample name

PBAT

PBAT/

m-PPZn-01

PBAT/

m-PPZn-05

PBAT/

m-PPZn-10

0.286±0.002

0.266±0.002

0.246±0.002

0.235±0.003

There are some detailed concerns:

  1. Line 50; The authors stated that “Considering both PPZn and MMT are two-dimensional layered inorganic material, the feature on UV photodegradation of polymer/PPZn composites attract our attention.” This vague and weak motivation. The authors should motivate the study stronger in terms of literature review and studying underlying mechanisms.

Answer: The authors agree with the reviewer’s comment. The authors have added the novelty of this research in the Introduction section on page 2, line 50 “Since PPZn shows a broad UV absorption bands between 300 to 400 nm, thus, we are concerned with their effect on the UV photodegradation of biodegradable polymer/PPZn composites [27]. This study is the first to discuss the feasibility of PPZn to affect the photodegradation of PBAT.” according to the reviewer’s comment. One more reference has been added in this manuscript.

[27] Singleton, R.; Bye, J.; Dyson, J.; Baker, G.; Ranson, R.M.; Hix, G.B. Tailoring the photoluminescence properties of transition metal phosphonates. Dalton Transactions 2010, 39, 6024-6030, doi:10.1039/C000531B.

  1. The FTIR spectra are needed to be clarified in the way how are they normalized to a characteristic peak? (Figure 1). If not, that would be misleading.

Answer: The data of FTIR spectra recorded using a Spectrum One spectrometer (Perkin-Elmer, MA, USA) in the range from 450–4000 cm−1 was not normalized.

  1. The authors used relative intensity in the XRD Figure? Normalized to which peak?

Answer: The data of m-PPZn was normalized to the peak at 2θ = 3.30°. The data of PBAT composites were normalized to the peak at 2θ = 23.21°. The authors have added this sentence on page 5, line 133 ” The data of m-PPZn was normalized to the peak at 2θ = 3.30°. The data of PBAT and composites were normalized to the peak at 2θ = 23.21°.” according to the reviewer’s comment.

  1. The authors stated that (Line 136); “Moreover, the morphologies of PBAT/m-PPZn-10 was investigated by TEM, shown in Figure 4. The stacking structure of m-PPZn is clearly shown, which is good agreement with the WAXD data.” The TEM doesn’t show this conclusion?!!

Answer: The interlayer spacing of m-PPZn calculated using WAXD is 2.67 nm. The interlayer spacing of m-PPZn estimated using TEM is also close to this dimension. The TEM image is shown below.

  1. Figure 6 together with is too misleading. The authors need to assign RC=O and FC=O clearly in Figure 6. Then the authors need to include one of the nanocomposites before and after irradiation in the Figure. The results in Table 1 showed that the ratio (indicating the photodegradation) for all samples is almost in the same range. However, the authors stated that (Line 170) “The decrease in value of ratio with increasing m-PPZn content demonstrates that m-PPZn retards the photodegradation of PBAT.”, which is not true.

Answer: The authors have added the FTIR results of PBAT/m-PPZn composites in Figure 6 according to the reviewer’s comment. The data of carbonyl index and experimental error has been newly added in Table 1. From this result, it is clear that the carbonyl index is decreased with increasing the m-PPZn contents.

Table 1. Ratio of FCO to RCO of samples after 21 days UV irradiation.

Sample name

PBAT

PBAT/

m-PPZn-01

PBAT/

m-PPZn-05

PBAT/

m-PPZn-10

0.286±0.002

0.266±0.002

0.246±0.002

0.235±0.003

  1. Figure 10 is also misleading. Which molecular weight? Mn or Mw? How are the PDI and molecular weight distribution? The authors need to bring all information at least in supporting information.

Answer: The molecular weight shown in Figure 10 is number-average of molecular weight (Mn). Figure 10 has been modified according to the reviewer’s comment. The additional information of Mw and PDI was listed below according to the reviewer’s comment.

Table 1. Molecular weight (Mn, kDa) of PBAT and PBAT/m-PPZn composites after different UV irradiation day.

   Sample

name

UV irradiation day

PBAT

PBAT/

m-PPZn-01

PBAT/

m-PPZn-05

PBAT/

m-PPZn-10

0

30.595

30.086

30.571

31.079

7

10.191

10.918

13.214

15.070

14

7.021

8.294

10.801

12.149

21

6.617

7.582

9.337

9.752

Table 2. Molecular weight (Mw, kDa) of PBAT and PBAT/m-PPZn composites after different UV irradiation day.

   Sample

name

UV irradiation day

PBAT

PBAT/

m-PPZn-01

PBAT/

m-PPZn-05

PBAT/

m-PPZn-10

0

56.195

55.761

55.637

56.203

7

21.844

23.386

28.117

31.198

14

10.074

14.730

22.647

26.725

21

9.629

11.957

17.268

20.238

Table 3. Polydispersity index (PDI) of PBAT and PBAT/m-PPZn composites after different UV irradiation day.

   Sample

name

UV irradiation day

PBAT

PBAT/

m-PPZn-01

PBAT/

m-PPZn-05

PBAT/

m-PPZn-10

0

1.84

1.85

1.82

1.81

7

2.14

2.14

2.13

2.07

14

1.43

1.78

2.10

2.20

21

1.44

1.58

1.85

2.08

  1. The authors need to discuss the kinetics of photodegradation.

Answer: The authors would like to consider the reviewer’s suggestion by designing the kinetics experiments for the further investigations.

Reviewer 2 Report

The subject matter could be interesting for the general readership of the journal, however, I feel that the manuscript needs major improvement. I would suggest the authors to consider following recommendations carefully and resubmit their manuscript after through revision: 1. One of the major lacking of the presented work is a clear statement on the novelty of the research. Please discuss these in the Introduction section and state what is new in this study. 2. “PPZn were prepared by a similar approach reported previously” – Although authors referred to a previous communication, please provide some additional information on the sample preparation like % of phenylphosphonic acid and zinc nitrate hexahydrate in water. 3. “The mixing solution was dried in vacuum at 40 °C for 3 days.” – Please correct the sentence: either “mixed solution” or “solution mixture” 4. “These composites and neat PBAT were hot pressed and cut into 1×4 cm pieces.” – Please mention the type of device used for hot press. What were the processing parameters/conditions? 5. “The samples were irradiated under an UV lamp” – Please mention the wavelength. 6. “The PBAT and its composites were performed in attenuated total reflection (ATR) mod.” – Do you mean “mode”? 7. “The UV spectrum experiments was acquired” – Do you mean “UV-Vis spectra were acquired”? 8. “Figure 2. shows the UV–Vis reflectance spectrum of m-PPZn.” – Please include the spectra of control samples and compare them. 9. “Figure 3 shows the WAXD result of the m-PPZn, PBAT, and PBAT/m-PPZn composites.” – Authors stated the observations only without any proper scientific insight. What did we learn from these spectra - from the peak positions and the interlayer spacing? Please discuss in the text. 10. “By increasing m-PPZn content, the appearing peak positions of m-PPZn remained unchanged, and the intensities of these peaks increased.” – Did the authors normalized those peaks? Otherwise this statement is not meaningful. Please discuss in the text. 11. “Similar FTIR results of PBAT/m-PPZn composites were also observed.” – Please include those spectra in Fig. 6 and discuss in details. 12. How was it possible to have exactly same molecular weight of all four samples (Fig. 10, 0 day)? 13. - The English language of this manuscript needs improvement. I have mentioned some corrections in my comments, however, there are many more in the manuscript. I am recommending a proofread of the manuscript thoroughly using a language editing service.

Author Response

Response to the comments of reviewers

The authors appreciate the referees’ careful reading and thoughtful suggestions. Points by point responses to reviewer’s comments are discussed below.

Reviewer #2:

The subject matter could be interesting for the general readership of the journal, however, I feel that the manuscript needs major improvement. I would suggest the authors to consider following recommendations carefully and resubmit their manuscript after through revision:

  1. One of the major lacking of the presented work is a clear statement on the novelty of the research. Please discuss these in the Introduction section and state what is new in this study.

Answer: The authors agree with the reviewer’s comment. The authors have added the novelty of this research in the Introduction section on page 2, line 50 “Since PPZn shows a broad UV absorption bands between 300 to 400 nm, thus, we are concerned with their effect on the UV photodegradation of biodegradable polymer/PPZn composites [27]. This study is the first to discuss the feasibility of PPZn to affect the photodegradation of PBAT.” according to the reviewer’s comment. One more reference has been added in this manuscript.

[27] Singleton, R.; Bye, J.; Dyson, J.; Baker, G.; Ranson, R.M.; Hix, G.B. Tailoring the photoluminescence properties of transition metal phosphonates. Dalton Transactions 2010, 39, 6024-6030, doi:10.1039/C000531B.

  1. “PPZn were prepared by a similar approach reported previously” – Although authors referred to a previous communication, please provide some additional information on the sample preparation like % of phenylphosphonic acid and zinc nitrate hexahydrate in water.

Answer: The authors agree with the reviewer’s comment. The authors have added the preparation of PPZn in the Materials and Methods on page 2, line 70 “Typically, the 1 g phenylphosphonic acid and 1.88 g zinc nitrate hexahydrate were individually dissolved in 40 ml and 20 ml deionized water then mixed together followed by the addition of 0.1 M aqueous sodium hydroxide to reach pH 5–6.” according to the reviewer’s comment.

  1. “The mixing solution was dried in vacuum at 40 °C for 3 days.” – Please correct the sentence: either “mixed solution” or “solution mixture”.

Answer: The authors have corrected the sentence in the Materials and Methods on page 2, line 79 “The mixed solution was dried in vacuum at 40 °C for 3 days.” according to the reviewer’s comment.

  1. “These composites and neat PBAT were hot pressed and cut into 1×4 cm pieces.” – Please mention the type of device used for hot press. What were the processing parameters/conditions?

Answer: The authors have added additional information on the processing parameters/conditions in the Materials and Methods on page 2, line 81 “These composites and neat PBAT were hot-pressed using a hot-press machine under the pressure of 10 kgf/cm2 at 120 °C for 10 minutes. The samples were cut into 1×4 cm pieces for further investigation.” according to the reviewer’s comment.

  1. “The samples were irradiated under an UV lamp” – Please mention the wavelength.

Answer: The authors have added additional information on the wavelength in the Materials and Methods on page 2, line 85 “The samples were irradiated under an UV lamp (Philips CLEO HPA 400S, Amsterdam, Netherlands) with a mainly radiation between 300 and 400 nm.” according to the reviewer’s comment.

  1. “The PBAT and its composites were performed in attenuated total reflection (ATR) mod.” – Do you mean “mode”

Answer: The authors have corrected the misspelling in the Materials and Methods on page 3, line 90 “The PBAT and its composites were performed in attenuated total reflection (ATR) mode.” according to the reviewer’s comment.

  1. “The UV spectrum experiments was acquired” – Do you mean “UV-Vis spectra were acquired”?

Answer: The authors have corrected the mistake in the Materials and Methods on page 3, line 93 “UV-Vis spectra were acquired by a U-3900 UV–Vis spectrophotometer (Hitachi, Tokyo, Japan) in the range of wavelength from 250 to 400 nm.” according to the reviewer’s comment.

  1. “Figure 2. shows the UV–Vis reflectance spectrum of m-PPZn.” – Please include the spectra of control samples and compare them.

Answer: The authors have added the UV-Vis spectrum of PPZn in Figure 2 according to the reviewer’s comment. The comparison of UV-Vis spectrum has been added on page 4, line 126 “Figure 2 shows the UV–Vis reflectance spectrum of the PPZn and m-PPZn. In the UV-A and UV-B regions, more than 80% of the UV light is reflected, indicating the barrier ability of the PPZn. A lower reflectance in the UV-C region is attributed to the absorption via the π→π* transition of the phenylphosphonate anion [27]. In the region of UV-A and UV-B, the organic modifier hexadecylamine was slightly reduced the reflectance of the PPZn.” according to the reviewer’s comment.

  1. “Figure 3 shows the WAXD result of the m-PPZn, PBAT, and PBAT/m-PPZn composites.” – Authors stated the observations only without any proper scientific insight. What did we learn from these spectra - from the peak positions and the interlayer spacing? Please discuss in the text.

Answer: The authors have added more discussion on the WAXD results on page 5, line 147 “These results indicate that the initial stacking structure of the m-PPZn is retained in the PBAT/m-PPZn composites, and the addition of the m-PPZn does not change the crystal structure of the PBAT.” according to the reviewer’s comment.

  1. “By increasing m-PPZn content, the appearing peak positions of m-PPZn remained unchanged, and the intensities of these peaks increased.” – Did the authors normalized those peaks? Otherwise this statement is not meaningful. Please discuss in the text.

Answer: The data of m-PPZn was normalized to the peak at 2θ = 3.30°. The data of PBAT composites were normalized to the peak at 2θ = 23.21°. The authors have added this sentence on page 5, line 135 ” The data of m-PPZn was normalized to the peak at 2θ = 3.30°. The data of PBAT and composites were normalized to the peak at 2θ = 23.21°.” according to the reviewer’s comment.

  1. “Similar FTIR results of PBAT/m-PPZn composites were also observed.” – Please include those spectra in Fig. 6 and discuss in details.

Answer: The authors have added the FTIR results of PBAT/m-PPZn composites in Figure 6 according to the reviewer’s comment. The comparison of the FTIR results of PBAT/m-PPZn composites has been added on page 7, line 172 “Figure 6 shows the FTIR spectrum of PBAT and the PBAT/m-PPZn composites before and after 21 days of UV irradiation. Before the UV irradiation, the FTIR spectrum of PBAT and the PBAT/m-PPZn composites showed an absorbance peak at 1721 cm−1 which is attributed to the C=O group (Rc=o). The peaks at 1268, 1250, 1166, 1118, and 1103 cm−1 were attributed to the ester group. The peaks at 720 cm−1 were attributed to the methylene groups, while that at 1505 and 875 cm−1 were attributed to the benzene ring [38,39]. The intensity of these peaks became insignificant after irradiation, which indicates the chemical decomposition of the PBAT via photodegradation. In addition, a broad carbonyl peak (1850–1550 cm−1) was observed, which indicates the chain scission of the ester group. The left shoulder (1850–1750 cm−1) of the carbonyl peak indicates the formation of the free CO (FCO), while the right shoulder (1670–1550 cm−1) indicates the formation of a lower molecular weight ester (LMWE) [10]. The chain scission at the ester linkages of the PBAT from the photodegradation via Norrish I process since the carbonyl group is easily photodegraded [10,38].” according to the reviewer’s comment.

  1. How was it possible to have exactly same molecular weight of all four samples (Fig. 10, 0 day)?

Answer: The authors have newly measured the molecular weight of all samples and the data has been added in Figure 10 according to the reviewer’s comment.

  1. - The English language of this manuscript needs improvement. I have mentioned some corrections in my comments, however, there are many more in the manuscript.

I am recommending a proofread of the manuscript thoroughly using a language editing service. 

Answer: The writing of this manuscript in English has been edited by Enago, Crimson Interactive Pvt. Ltd. The certificate of editing is attached below.

Round 2

Reviewer 1 Report

I still think that the research is not designed well based on the claim that the authors made in the introduction. Some of the results are still misleading (FTIR and XRD). I suggested having some kinetics results, which might help both deep understanding and confirmation of photodegradation. The authors added the error level, which is really low. This is really vague, how they calculate that? Several IR spectra from one sample or several samples? Is it coming from deconvolution? A simple deconvolution also has a high level of error based on different function and picking base line, and etc.

I regret to say that this is not worthy for publication.

Reviewer 2 Report

Acceptance is recommended.